# Clinical Characteristics and Outcomes of Extrapulmonary Nontuberculous Mycobacterial Infections in a Tertiary-Care Hospital: A Retrospective Study

**DOI:** 10.3390/jcm13154373

**Published:** 2024-07-26

**Authors:** Seulki Kim, A Reum Kim, Moonsuk Bae, Seungjin Lim, Su Jin Lee

**Affiliations:** 1Division of Infectious Diseases, Department of Internal Medicine, Pusan National University Yangsan Hospital, Yangsan 50612, Republic of Korea; seul2968@naver.com (S.K.); ar6493@naver.com (A.R.K.); carukeion@gmail.com (M.B.); babopm@naver.com (S.L.); 2Research Institute for Convergence of Biomedical Science and Technology, Pusan National University Yangsan Hospital, Yangsan 50612, Republic of Korea; 3Department of Internal Medicine, Pusan National University School of Medicine, Pusan 46241, Republic of Korea

**Keywords:** nontuberculous mycobacteria, extrapulmonary nontuberculous mycobacterial infection, diagnosis, pathogen isolation, clinical outcome, treatment

## Abstract

**Background/Objectives:** The incidence of nontuberculous mycobacterial (NTM) infections has increased globally; however, the clinical manifestations and optimal treatment strategies for extrapulmonary NTM infections remain poorly defined. This study assessed the clinical manifestations and treatment outcomes of extrapulmonary NTM infections. **Methods:** Data from adult patients with suspected extrapulmonary NTM infections at a tertiary-care hospital from 2009–2022 were categorized into NTM disease and isolation groups. Diagnosis of NTM disease relied on stringent criteria, whereas isolation required NTM isolation without meeting the criteria for infection. **Results:** Among 75 patients evaluated, 32 (42%) were diagnosed with NTM disease and 43 (57%) with NTM isolation. History of immunosuppressant use within the past 3 months (*p* = 0.070) and injection (*p* = 0.001) were more frequent in the disease group. The median interval from symptom onset to evaluation was 106.6 and 20 days in the disease and isolation groups, respectively. The prevalence of positive NTM polymerase chain reaction results (36.4%, *p* = 0.003) and acid-fast bacillus staining (39.1%, *p* < 0.001) was significantly higher in the disease group than in the isolation group. *Mycobacterium intracellulare* (21.9%), *M. abscessus* (15.6%), *M. chelonae* (9.4%), and *M. fortuitum* complex (9.4%) were the most frequently identified species. Of the 27 patients in the disease group who received treatment, 13 improved, four experienced treatment failure, seven were lost to follow-up, and three died during treatment, with one death directly attributable to NTM disease. **Conclusions:** NTM disease exhibits a spectrum of clinical manifestations. Accurate diagnosis is crucial for initiating effective treatment.

## 1. Introduction

Nontuberculous mycobacteria (NTM) are commonly found in the environment but rarely lead to disease [1,2]. Although less harmful than mycobacterium tuberculosis, NTM can cause illness, particularly in individuals with compromised or overreactive immune systems [3,4]. Although NTM infections most frequently affect the lung [5], approximately 25% of cases of NTM infection occur at extrapulmonary sites, such as the lymph nodes, skin, and, occasionally, bones [2,5,6,7,8].

In recent years, the incidence of NTM infections has increased globally, thereby highlighting the growing significance of this group of microorganisms as opportunistic pathogens [9,10,11]. However, this trend has increased the need for comprehensive epidemiological research owing to the prevalent uncertainties regarding optimal treatment modalities, treatment duration tailored to different NTM species, and the role of surgical intervention in managing these infections [12,13]. The difference in the trends of pulmonary and extrapulmonary NTM infection suggests that these two conditions represent different diseases with different risk factors [5]. Pulmonary NTM disease typically occurs in individuals with pre-existing structural lung damage and is associated with aging [5]. Conversely, extrapulmonary NTM disease occurs in immunocompromised individuals or post-NTM invasion into damaged tissues [2,5].

Moreover, extrapulmonary NTM diseases are caused by a diverse range of bacterial pathogens. Culture and molecular techniques are used to differentiate NTM. As identification of subspecies and drug susceptibility profiling are critical factors in selecting appropriate treatment of extrapulmonary NTM diseases, treatment initiation and initiation of targeted treatment specific to the causative pathogen are often delayed. To address critical gaps in knowledge, we conducted an extensive study on a diverse cohort of adult patients with extrapulmonary NTM infections. Collectively, we believe that our findings will help deepen understanding of extrapulmonary NTM infections and provide healthcare professionals with guidance on diagnosing, managing, and treating these infections, thereby improving patient outcomes and optimizing healthcare strategies.

## 2. Materials and Methods

### 2.1. Study Design and Data Collection

The study utilized a retrospective cohort design. We included all adult patients (age ≥ 18 years) admitted to Pusan National University Yangsan Hospital from 2009 to 2022, who were confirmed to have extrapulmonary NTM infections through rigorous culture-based diagnostic methods. Pusan National University Yangsan Hospital, a 1300-bed, university-affiliated tertiary-care teaching hospital located in Gyeongsangnam-do, Republic of Korea, provided a comprehensive setting for this analysis. Information regarding demographics, pre-existing medical conditions, symptoms and signs at presentation, microbiological and imaging findings, type and duration of antimicrobial therapy, and patient outcomes were meticulously extracted from the medical records.

Distinguishing between pathogen, concomitant infection, and saprophytic isolates of extrapulmonary NTM is challenging [7]; therefore, we divided the patients into two distinct groups: NTM disease and NTM isolation.

NTM disease: Diagnosed when any of the following criteria are met:Positive cultures from at least two separate samples, confirming the consistent presence of NTM.Biopsy findings indicating mycobacterial histopathologic features, such as granulomatous inflammation or presence of acid-fast bacilli (AFB), confirmed through NTM polymerase chain reaction (PCR).Positive culture from normally sterile extrapulmonary sites, indicating active infection.

NTM isolation: Defined as the presence of NTM in non-sterile sites without sufficient clinical, radiological, or histopathologic evidence of active disease. This includes single positive cultures from non-sterile sites or cases where the evidence suggests colonization or contamination rather than an active infection.

### 2.2. NTM Testing Methods

Culture: Clinical specimens were cultured on BACTEC MGIT 960 system (Becton Dickinson, Franklin Lakes, NJ, USA; MGIT).

Acid-fast bacilli (AFB) staining: Pathological specimens were stained using the Ziehl–Neelsen method to detect AFB microscopically. Other specimen types were stained for AFBs using KS-S100 (KS Co. Ltd., Naju, Republic of Korea) staining solution (fluorochrome method).

NTM PCR: When NTM infection was clinically suspected, specimens were subjected to PCR testing for rapid identification of mycobacterial DNA. This molecular diagnostic method is useful for rapid and accurate detection of NTM.

Line probe assay (LPA): Positive culture isolates were sent to the Korean Tuberculosis Association for species identification using the LPA. This method allowed for the differentiation of NTM species and enabled antimicrobial therapy to be tailored appropriately.

### 2.3. Ethics Statement

This study was approved by the Institutional Review Board (IRB) of the Pusan National University Yangsan Hospital (IRB no. 55-2024-011). The requirement for informed consent was waived owing to the retrospective nature of the study.

### 2.4. Definitions of Terms

Immunosuppressant therapy was defined as the use of corticosteroids (the equivalent of ≥0.3 mg/kg/day of prednisone) for at least 3 weeks or other recognized T-cell immunosuppressants such as tumor necrosis factor-α blockers and calcineurin inhibitors within 90 days prior to the hospital visit.

The term “history of injection” was used to refer to a previous injection at the site at which the NTM infection was later identified. This encompassed all forms of injection, including intramuscular, subcutaneous, or intravenous injection, that might have occurred prior to the diagnosis of the NTM infection. The term “history of surgery” was used to refer to prior surgical procedures performed at the site where the NTM infection was identified.

### 2.5. Statistical Analyses

The clinical and laboratory test results of the NTM disease and NTM isolation groups were compared. Continuous data were described as medians and interquartile ranges (IQRs), and the Mann–Whitney U test was used to compare continuous variables between groups. Fisher’s exact test was used to compare categorical variables between groups. All tests were two-tailed, and differences with *p* values < 0.05 were considered statistically significant. Statistical analyses were performed using SPSS (version 23.0; IBM Corp., Armonk, NY, USA).

## 3. Results

### 3.1. Patient Characteristics

In this study, 75 patients with culture-proven NTM were enrolled (Figure 1). Among these, 32 (42%) and 43 (57%) were diagnosed with NTM disease and isolations, respectively. The clinical characteristics of the groups are presented in Table 1. The median age was 64.0 (IQR, 51.5–74) years, with no significant difference between the NTM disease and NTM isolation group (68.5 [IQR 52.8–75.0] years vs. 63.0 [IQR 48.5–72.5] years; *p* = 0.379).

Eleven of the 75 patients (15%) had autoimmune diseases. The prevalence of autoimmune disease was higher in the NTM disease group (6 of 32; 19%) than in the isolation group (5 of 43; 12%), but this difference was not statistically significant (*p* = 0.071). The history of immunosuppressant medication use within the past 3 months was also higher in the NTM disease group (9 of 32; 28%) than in the NTM isolation group (5 of 43; 12%), but this difference was also not statistically significant (*p* = 0.070). Steroid use was higher in the NTM disease group than in the NTM isolation group, but this difference was not statistically significant (*p* = 0.067).

Compared with patients in the NTM isolation group, patients in the NTM disease group were significantly more likely to have a history of injection (9 of 32, 28% vs. 1 of 43, 2%; *p* = 0.001). However, the history of previous surgery at the site where the NTM was identified did not differ significantly between the groups (*p* = 0.840).

Five patients in the isolation group (12%) and no patients in the NTM disease group had a history of tuberculosis (*p* = 0.067). In the NTM disease group, 3 of 32 patients (9%) had co-infections with lung NTM.

### 3.2. Site of NTM Isolation

The prevalence of NTM isolates in the musculoskeletal system was higher in patients with NTM disease (17 patients, 53%) than in patients with NTM isolations (15 patients, 35%). Specifically, infections were identified in muscles (e.g., thigh, calf, paravertebral), bones (e.g., femur, tibia, spine), and joints (e.g., knee, shoulder). Muscle infection accounted for 22% of cases, bone infections for 19%, and joint infections for 12%. However, MTM disease versus isolation did not differ significantly by site (*p* = 0.114) (Table 2). The distribution of NTM isolates in the skin and soft tissue was similar between the NTM disease (13 patients, 41%) and NTM isolation (16 patients, 37%) groups. NTM isolates from the gastrointestinal (GI) mucosa were more prevalent in the isolation group (five patients, 12%) than in the NTM disease group (0 patients); however, this difference was not statistically significant (*p* = 0.067). NTM isolates from the lymph nodes, urine, and cerebrospinal fluid (CSF) were infrequent in both groups.

Four patients in the NTM disease group (13%) had bacteremia, of whom one died before the evaluation had been completed, and three had spread of NTM infection from a localized site to the bloodstream. One patient in the NTM infection group had an infection in an arteriovenous graft.

### 3.3. Clinical Manifestation of the Patients with NTM Disease or Isolation

The main clinical symptom among the patients was pain (Appendix A), reported by 21 patients (66%) in the NTM disease group and 27 patients (63%) in the NTM isolation group (*p* = 0.800). Fever was observed in five patients (16%) in the NTM disease group and nine patients (21%) in the NTM isolation group (*p* = 0.560). The median interval from symptom onset to evaluation was longer in the NTM disease group (106.6 days) than in the isolation group (20 days), but this difference was not statistically significant (*p* = 0.350).

Skin lesions such as papules, nodules, patches, ulcers, and masses were more significantly common in the disease group (seven patients, 22%) than in the isolation group (one patient, 2%; *p* = 0.009). The locations of skin manifestations in NTM disease were precisely identified. The skin lesions were located on the legs in three patients (43%), trunk in three patients (43%), and face in one patient (14%). Among the patients with skin lesions on the trunk, one patient each had lesions on the abdomen, chest wall, and back. Among the seven patients with skin lesions, three had lesions associated with previous surgical sites, and two had a history of injections at the affected site. The characteristics of the skin lesions were variable, with papules and nodules in two patients (29%), extensive patches in two patients (29%), ulcers in one patient (14%), masses in one patient (14%), and wound dehiscence in one patient (14%).

Other nonspecific symptoms, such as cellulitis-like lesions, graft stenosis, dyspnea, and general weakness, were reported in varying proportions in both groups, with no significant differences.

### 3.4. Initial Laboratory Findings of Patients with NTM Disease

Laboratory tests from patients with NTM diseases showed a mild increase in C-reactive protein (CRP) levels compared with the normal range (Appendix A). No differences were observed in the complete blood count and differential count, and no specific findings were identified that could confirm the presence of NTM. These findings suggest that although most hematologic parameters (white blood cell count, neutrophil count, lymphocyte count, platelet count, and hemoglobin) remain within reference ranges, a mild elevation in CRP levels is a sign of inflammation. These results, combined with other clinical and diagnostic criteria, may assist in diagnosing NTM disease. Notably, clinicians should interpret these results within the context of the clinical presentation of each patient and additional diagnostic tests.

### 3.5. Comparison of Diagnostic Testing Results

A significantly higher proportion of patients in the disease group (68.8%) underwent NTM PCR testing compared with those in the isolation group (18.6%). Among those tested, 8 out of 22 NTM PCR results were positive, indicating the presence of NTM (Table 3). The performance of AFB staining was similar in both groups, with 71.9% in the disease group and 86.0% in the isolation group having samples tested using AFB staining. In the NTM disease group, 39.1% had a positive AFB stain result, compared with none in the NTM isolation group (*p* < 0.001).

The NTM species was identified in 78% of the disease group, but in the isolation group, 91% of isolates could not be identified at the species level. Specifically, in the NTM disease group, the most common species identified were *Mycobacterium intracellulare* (22%), and *Mycobacterium abscessus* (16%). Other species identified included *Mycobacterium chelonae*, *Mycobacterium fortuitum* complex, *Mycobacterium massilense*, *Mycobacterium kansasii*, *Mycobacterium marinum* or *Mycobacterium ulcerans*, and *Mycobacterium avium*. Amplification failure occurred in the specimen from one patient in the isolation group.

Rapidly growing mycobacteria (RGM) were the most common causes of skin, soft tissue, bone, and joint infections (Figure 2 and Figure 3). Disseminated infection was uncommon, but most disseminated infections were caused by RGM.

### 3.6. Treatment and Clinical Outcomes

In the disease group, 66% of patients underwent NTM drug susceptibility testing, whereas in the isolation group, only one patient (2%) underwent NTM drug susceptibility testing (Table 4) (*p* < 0.001). Of the 21 patients who underwent drug susceptibility testing, three (14%) had clarithromycin-resistant infections, and 17 (81%) had quinolone-resistant infections.

Of the 27 patients who received antimicrobial therapy for NTM infection, six (22%) received clarithromycin monotherapy, and the majority (67%) were treated with combination therapy that included clarithromycin. Other agents, including antitubercular agents (2 patients, 7%), amikacin (7 patients, 26%), and carbapenem (6 patients, 22%), were used less frequently.

Of the 32 patients with NTM disease, 27 (84%) started treatment. Of the five patients who did not initiate treatment, four were lost to follow-up before treatment initiation or declined treatment, and one died of comorbidities before treatment could be initiated. The median total duration of antibiotic treatment was 240 days (range: 115–378 days). The median duration of follow-up for patients was 319 days (range: 176.5–812.5 days). During treatment, 7 out of 27 patients (26%) were lost to follow-up, whereas 13 patients (48%) showed clinical improvement. Four patients (15%) experienced treatment failure, and three patients (11%) died during the course of treatment, with one death attributed to NTM disease, and two attributed to other conditions. The median duration of hospitalization was 26.0 days (IQR: 8.0–60.5 days).

## 4. Discussion

In this study, we highlight the complexity of diagnosing and managing NTM infections. Among the 75 patients with culture-confirmed NTM, 32 were diagnosed with NTM disease and 43 with NTM isolation. Although we observed no significant differences in underlying diseases and NTM infections, the prevalence of autoimmune diseases and steroid use was higher in those with NTM disease. This suggests that the use of steroids might be a risk factor for NTM infection, especially in the context of immunosuppression, which is associated with an increased risk of extrapulmonary NTM infection [2,14].

Skin and soft tissue infections are often preceded by posttraumatic wound infection or procedures [14,15]. Our findings indicate that a history of injection is a significant risk factor for NTM infection. This underscores the importance of thorough patient evaluation, including obtaining a detailed medical history, to identify risk factors and facilitate early diagnosis.

According to a study conducted in four tertiary hospitals in Korea, skin and soft tissue infections are the most common forms of extrapulmonary NTM infection, followed by bone and joint infections [16]. In the present study, extrapulmonary NTM infections were frequently observed in musculoskeletal systems and in the skin and soft tissues. Osteomyelitis and arthritis were the most common types of infections observed, with skin infections following closely behind.

The distinction between NTM isolation and NTM disease is crucial. NTM in the blood or associated with vascular access sites indicates systemic infection, necessitating more aggressive treatment. The fatal case of NTM bacteremia highlights the severity of bloodstream infections.

We also recognized that not all NTM isolations are indicative of active infection. NTM isolates in urine, ascites, and GI mucosa were not regarded as major pathogens. In the cases of ascites, NTM was not confirmed in follow-up paracentesis, and there was no evidence of peritonitis. Cases of NTM isolates in CSF and lymph nodes were confirmed during treatment for disseminated tuberculosis and were considered to be contaminants. Each case requires individual evaluation based on clinical context and symptoms to determine the appropriate course of action and provide the most suitable care and treatment.

The clinical manifestations of NTM infection vary, with pain and skin lesions being common manifestations. The longer duration of symptoms in the disease group suggests the importance of early diagnosis and treatment. Although some symptoms, such as skin lesions, may be more prevalent in NTM disease, many other symptoms, including fever and pain, did not differ significantly between groups.

Diagnostic tests play a vital role in confirming NTM diseases and guiding treatment decisions. NTM PCR testing was significantly more likely to yield positive results in patients with NTM disease. Additionally, a positive AFB stain result indicating the presence of mycobacteria, was more common in the disease group. These findings highlight the importance of comprehensive diagnostic evaluations, including NTM PCR and AFB staining, in distinguishing between NTM diseases and isolations.

According to a previous study, the number of NTM isolation cases is increasing in Korea, and the most common extrapulmonary NTM species identified were *M. intracellulare*, *M. avium*, *M. kansasii*, *M. abscessus*, *M. fortuitum* complex, *M. chelonae*, and *M. ulcerans* [17]. It was about isolated NTM and not about pathogens. Region-specific differences have also been reported; in Gyeonsang-do, the predominant NTM species identified, in order of decreasing frequency, included M. *intracellulare*, *M. abscessus*, *M. fortuitum* complex, and *M. kansasii* [17]. This distribution partly aligns with our findings. In our study, *M. intracellulare* was the most common species, although the ranking of other species differed. Another study on pathogens causing extrapulmonary NTM disease, conducted from January 2006 to June 2018 at four tertiary hospitals in Korea, reported the following distribution: *M. intracellulare*, *M. fortuitum* complex, *M. abscessus*, *M. massilense*, *M. chelonae*, *M. kansasii*, *M. avium*, *M. ulcerans*, and *M. marinum* [16]. In our study, we identified *M. intracellulare*, *M. abscessus*, *M. chelonae*, *M. fortuitum complex*, *M. massilense*, *M. kansasii*, *M. marinum*, *M. ulcerans*, and *M. avium*. Specific species, such as *M. abscessus*, were found exclusively in the disease group, whereas species, such as *M. avium*, were infrequent. Differences in species identification did not differ significantly between the NTM disease and NTM isolate groups for most species.

The identification of NTM species is crucial for understanding disease pathogenesis and making appropriate treatment choices. However, in this study, the NTM species was identified in only 78% of patients diagnosed with NTM disease, and overall 91% of NTM isolates did not have species-level identification because molecular diagnostic testing such as the LPA, was not routinely performed unless NTM disease was strongly suspected and further testing was requested by the treating clinician. This limitation should be considered when interpreting our findings. A high index of suspicion and clinical microbiological support are required for the accurate diagnosis of extrapulmonary NTM disease.

Drug susceptibility testing is crucial for appropriate tailoring of treatment regimens owing to the high prevalence of resistance to common antibiotics. Treating NTM disease is complex and often requires prolonged treatment with a combination of antibiotics tailored to the drug susceptibility profile of the patient. Monitoring and follow-up are essential owing to the possibility of treatment failure or complications. Challenges such as patient follow-up and adherence to treatment must be addressed to improve outcomes.

However, this study has some limitations. It covers a broad range of extrapulmonary NTM infections, caused by different NTM species at different anatomical sites, and the imaging findings were frequently unclear or inconclusive. Therefore, we were unable to include imaging findings in our analysis of the clinical characteristics. The relatively small sample size, particularly in patients diagnosed with NTM disease, might affect the generalizability of the findings and the ability to draw broader conclusions about NTM disease. Furthermore, enrolling patients from a tertiary-care hospital might exclude those with mild disease. Of the 75 patients initially suspected to have NTM infection, 43 (57%) did not have infection confirmed based on the disease definition, possibly introducing selection bias. This study was a retrospective study, which has inherent limitations, including potential biases and the inability to establish causality.

We acknowledge the limitations of our study, including the small sample size and the inclusion of diverse NTM species, which may lead to variability in treatment outcomes. The retrospective nature of the study further limits the generalizability of our findings. Different NTM species necessitate different treatment regimens, and the heterogeneity of our patient group may have impacted the overall conclusions. Despite these limitations, our study provides valuable preliminary insights into the clinical presentation and management of NTM diseases, underscoring the need for larger, prospective studies to validate these findings.

This study also provides valuable insights into the clinical characteristics and treatment outcomes of extrapulmonary NTM infections. Retrospective analyses are useful for analyzing real-world clinical data and identifying trends and associations that can inform future prospective research. We ensured the accuracy and reliability of the data by rigorously reviewing patient records and adhering to standardized diagnostic criteria.

## 5. Conclusions

Despite the limitations of our study, including the small sample size and retrospective single-center design, our findings suggest that comprehensive evaluations and diagnostic tests are crucial for accurately diagnosing extrapulmonary NTM disease. Our research makes significant contributions by providing detailed case analyses and addressing specific clinical challenges in this subset of NTM infections. Future research involving larger, multicenter cohorts is necessary to confirm these observations and further refine diagnostic and therapeutic strategies. Healthcare professionals should evaluate symptoms and medical history thoroughly and perform the tests necessary for accurate diagnosis.

Clinicians should suspect NTM infection in patients with chronic, non-healing skin lesions, particularly those with relevant exposure history or immunocompromised status. Early diagnosis and appropriate treatment are critical for effective management. Further studies are needed to refine treatment strategies and enhance understanding of extrapulmonary NTM infections, thus improving patient care and outcomes.

## Figures and Tables

**Figure 1 jcm-13-04373-f001:**
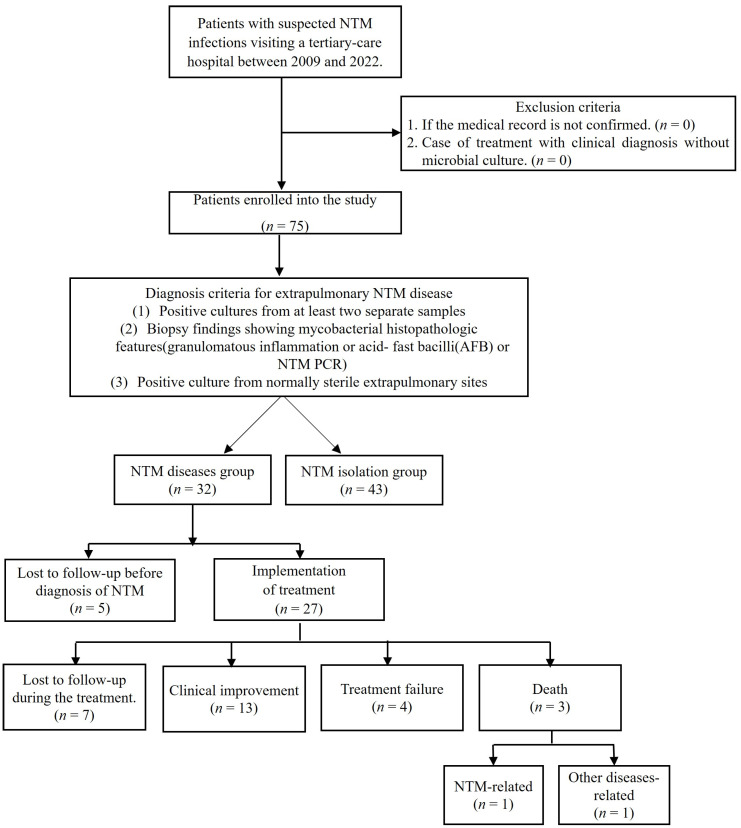
Flow chart of the patient selection process. Abbreviations: nontuberculous mycobacteria (NTM); acid-fast bacilli (AFB); polymerase chain reaction (PCR).

**Figure 2 jcm-13-04373-f002:**
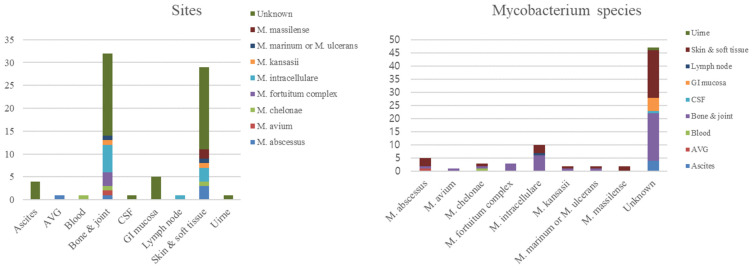
Species and site of infection of nontuberculous mycobacteria isolation. Abbreviations: GI, gastrointestinal; CSF, cerebrospinal fluid; AVG, arteriovenous graft.

**Figure 3 jcm-13-04373-f003:**
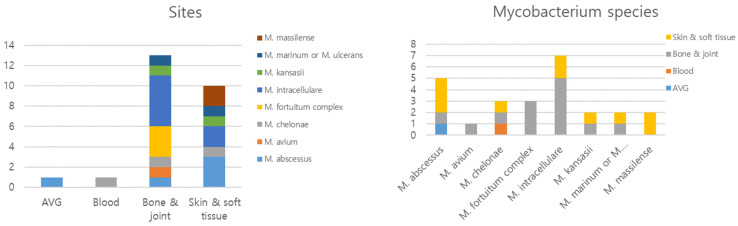
Species and site of infection of nontuberculous mycobacteria disease. Abbreviation: AVG, arteriovenous graft.

**Table 1 jcm-13-04373-t001:** Demographic of 75 patients suspected of having NTM.

Characteristic	All Patients (N = 75)	Disease (N = 32)	Isolation (N = 43)	*p* Value ^1^
Age, median (IQR), years	64.0 (51.5–74)	68.5 (52.8–75)	63.0 (48.5–72.5)	0.379
Male sex	42 (56)	14 (44)	28 (65)	0.065
Comorbidity				
Diabetes	23 (31)	8 (25)	15 (35)	0.359
Chronic kidney disease	11 (15)	3 (9)	8 (19)	0.264
Autoimmune disease	11 (15)	6 (19)	5 (12)	0.071
Malignancy	10 (13)	5 (16)	5 (12)	0.615
Cardiovascular disease	10 (13)	4 (12.5)	6 (14)	0.855
Chronic liver disease	7 (9)	2 (6)	5 (12)	0.692
Solid organ transplant	4 (5)	2 (6)	2 (5)	>0.999
Chronic lung disease	4 (5)	1 (3)	3 (7)	>0.999
History of immunosuppressant medication within 3 months	14 (19)	9 (28)	5 (12)	0.070
Steroid use	12 (16)	8 (25)	4 (9)	0.067
Cyclosporin	3 (4)	0	3 (7)	0.256
Methotrexate	3 (4)	2 (6)	1 (2)	0.572
Mycophenolic acid	3 (4)	2 (6)	1 (2)	0.572
Tacrolimus	2 (3)	2 (6)	0	0.179
Azathioprine	2 (3)	1 (3)	1 (2)	>0.999
Everolimus	1 (1)	0	1 (2)	>0.999
History of injection ^2^	10 (13)	9 (28)	1 (2)	0.001
History of surgery ^3^	11 (15)	5 (16)	6 (14)	0.840
Tuberculosis	5 (7)	0	5 (12)	0.067
Lung NTM	4 (5)	3 (9)	1 (2)	0.307

Data are the number (%) of patients unless otherwise indicated. ^1^ Mann–Whitney U test or Fisher’s exact test. *p* values < 0.05 were considered to be significant. ^2^ History of injection was defined as history of previously injected site where the NTM was identified. ^3^ History of surgery refers to the previous surgery for site where the NTM was identified. Abbreviations: NTM, nontuberculous mycobacteria, IQR, interquartile range.

**Table 2 jcm-13-04373-t002:** Site of nontuberculous mycobacteria isolates and comparison between disease and isolation groups.

Site of NTM Isolate	Disease (N = 32)	Isolation (N = 43)	*p* Value ^1^
Musculoskeletal system	17 (53)	15 (35)	0.114
Skin and soft tissue	13 (41)	16 (37)	0.764
Gastrointestinal mucosa	0	5 (12)	0.067
Ascites	0	4 (9)	0.131
Lymph node	0	1 (2)	>0.999
Urine	0	1 (2)	>0.999
Cerebrospinal fluid	0	1 (2)	>0.999
Arteriovenous graft	1 (3)	0	0.427
Blood	1 (3)	0	0.427
Accompanied by NTM bacteremia	3 (9)	0	0.073

Data are the number (%) of patients unless otherwise indicated. ^1^ Mann–Whitney U test or Fisher’s exact test. *p* values < 0.05 were considered to be significant. Abbreviation: NTM, nontuberculous mycobacteria.

**Table 3 jcm-13-04373-t003:** Comparison of diagnostic testing results in disease and isolation groups.

	Disease (N = 32)	Isolation (N = 43)	*p* Value ^1^
NTM PCR ^2^	22 (69)	8 (19)	<0.001
PCR positive ^3^	8/22 (36)	1/8 (12.5)	0.003
AFB stain ^2^	23 (72)	37 (86)	0.129
AFB positive ^3^	9/23 (39)	0/37	<0.001
NTM genus without species identification	7 (22)	39 (91)	<0.001
*M. intracellulare*	7 (22)	3 (7)	0.060
*M. abscessus*	5 (16)	0	0.012
*M. chelonae*	3 (9)	0	0.073
*M. fortuitum* complex	3 (9)	0	0.073
*M. massilense*	2 (6)	0	0.179
*M. kansasii*	2 (6)	0	0.179
*M. marinum* or *M. ulcerans*	2 (6)	0	0.179
*M. avium*	1 (3)	0	0.427
Amplification failure	0	1 (2)	>0.999
NTM drug susceptibility test	21 (66)	1 (2)	<0.001

Data are the number (%) of patients unless otherwise indicated. ^1^ Mann–Whitney U test or Fisher’s exact test. *p* values < 0.05 were considered to be significant. ^2^ NTM PCR and AFB staining test frequencies are indicated. ^3^ Positive results from the tests performed are documented. Abbreviations: NTM, nontuberculous mycobacteria, PCR, polymerase chain reaction, AFB, acid-fast bacillus.

**Table 4 jcm-13-04373-t004:** Treatment and clinical outcomes.

	Value (N = 32)	% or IQR
NTM drug susceptibility test	21	66
Clarithromycin resistance	3	14
Quinolone resistance	17	81
Treatment		
Clarithromycin monotherapy	6	22
Clarithromycin-based combination therapy	18	67
Alternative or other treatment agents		
Antituberculous agent	2	7
Amikacin	7	26
Carbapenem	6	22
Outcomes		
Lost to follow-up before diagnosis of NTM	5	16
Implementation of treatment (N)	27	84
Total duration of antibiotics (days), median (IQR)	240	115–378
Duration of follow-up (days), median (IQR)	319	176.5–812.5
Lost to follow-up during the treatment	7/27	26
Clinical improvement among patients receiving treatment (n/N)	13/27	48
Treatment failure	4/27	15
Death (n/N)	3/27	11
NTM-related	1	33
Other disease-related	2	67
Duration of hospitalization (days), median (IQR)	26	8.0–60.5

Data are the number (%) of patients unless otherwise indicated. Abbreviations: NTM, nontuberculous mycobacteria, IQR, interquartile range.

## Data Availability

The data used in this study are available from the corresponding author on reasonable request.

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
