# Peer review of "Clinical Characteristics and Outcomes of Extrapulmonary Nontuberculous Mycobacterial Infections in a Tertiary-Care Hospital: A Retrospective Study"

_jcm, 2024, doi:10.3390/jcm13154373_

Round 1

Reviewer 1 Report

Comments and Suggestions for Authors

1.       Author retrospectively analysis the 78 patients with NTM diseases or NTM isolates from one hospital, firstly the sample was too small, and is not enough to make the conclusion regarding the clinical characteristics and outcome of extra-pulmonary NTM infection, secondly, the retrospective study limited the value of data from small sample research in one hospital.

2.       Authors mentioned that 32 patients were diagnosed as NTM diseases, but 27 patients received treatment, what was the reason?

3.        The definition of NTM isolation group is still vague in this manuscript. the one of the criteria of NTM disease:” biopsy findings indicative of mycobacterial histopathologic features such as 79 granulomatous inflammation or presence of acid-fast bacilli (AFB)”, this sentence did not define the detail type of NTM, which was only at the level of histopathological and AFB rather than molecular pathological level, therefore this condition is not enough to define NTM diseases.

4.       The third condition for NTM diseases: “culture positive from normally sterile extrapulmonary sites.” Which is also not enough to make the diagnosis of NTM disease because of lack of identification of NTM strain or sub-types.

5.       Author did not follow up the outcome of NTM isolation group, because it still have possibility to turn into NTM diseases or develop some other Bacteria infection.

6.       How did the authors judge the case of positive NTM test in CSF as NTM isolation group?

7.       Author failed to introduce the method of NTM test which is essential for the evidence of the study.

8.       Table 3 and Table 4 had limited value and less specificities for NTM diseases, because these patients had many comorbidities which itself have variable abnormal symptoms and laboratory tests.

Comments on the Quality of English Language

English expression is smooth.

Reviewer 2 Report

Comments and Suggestions for Authors

Neglected topic, with significant clinical relevance. I was unable to identify the sites of the focus from the manuscript: "musculoskeletal system" is far too general: musclers? (which one) - bones? joints? which one ?

Even the skin-related manifestations need location; not to mention the extent /site etc...

The title refers to clinical characteristics - images (CT, MRI, PET/Scan) are sine qua non a clinical decision making.

Extension of the manuscript is required into this direction.

On the other hand, good part of unnecessery and lung, empty commonplaces and generalizations in the Discussion / Conclusion chapters can be truncated. No need for longum et latum repetitions and regurgitations of "textbook wisdom."

What the manuscript needs, a guideline for the ordinary clinician:

when should he/she be suspicious, that the lesion in the skin

Round 2

Reviewer 1 Report

Comments and Suggestions for Authors

1.       Author did not still distinguish NTM diseases from NTM colonizers. From the statement from the third condition which was the one of the criteria for NTM diseases, how did author distinguish this condition from NTM isolation? And author also failed to define clearly the meaning of NTM isolation.

2.       In this retrospective study with limited number of cases, author included only 27 patients with NTM diseases, however, they had different types of NTM species, therefore may had different treatment outcome, which had big occasionality due to too small group of cases. Besides that, different species of NTM had different type of regimens. From this point, this manuscript had limited significant on its conclusion.

3.       There were other similar studies published with more big size of cases, therefore this study is lack of novelty.

Reviewer 2 Report

Comments and Suggestions for Authors

Thank you for your hard work, the result speaks for itself

Author Response

Thank you for your encouraging feedback and acknowledgment of our efforts. We are committed to addressing any remaining concerns and ensuring the clarity and robustness of our study. Your constructive feedback has been invaluable in refining our manuscript. We look forward to submitting the revised version and hope it meets the standards of the journal.

If there are any further suggestions or areas of focus you believe would strengthen the manuscript, please do not hesitate to let us know. We greatly appreciate your time and expertise in reviewing our work.